# Experiencing the COVID-19 pandemic as a homeless person in Chennai, India: An interpretative phenomenological analysis

**Laalithya Konduru**[1,2¤]*, **Nishant Das**[3]

1 Department of Community Medicine, Sri Jagannath Healthcare and Research Center, Dhanbad, Jharkhand, India, 2 College of Medicine and Public Health, Flinders University, Bedford Park, South Australia, Australia, 3 Department of Humanities and Social Sciences, Indian Institute of Technology (ISM), Dhanbad, Jharkhand, India

¤ Current address: OHCHR, United Nations, New York, New York, United States of America
* laalithya@gmail.com

**Data Availability Statement:** Data cannot be shared publicly because the data contains identifiable information, the free availability of which is against the privacy guaranteed to the participants at the time of enrollment in the study.

## Abstract

Persons experiencing homelessness (PEHs) have a higher risk of morbidity and mortality compared to the general population and are highly vulnerable during the coronavirus disease (COVID-19) pandemic. Understanding their experience of the pandemic is important for mitigating the effects of the pandemic. Accordingly, we conducted a qualitative study on their lived experiences during the COVID-19 pandemic. Semi-structured interviews were conducted in nine PEHs from Chennai, India, recruited at food stalls between September 14–25, 2020. Data were analyzed using interpretive phenomenological analysis. The participants shared their experiences of the COVID-19 pandemic, its impact on them, and their coping strategies. All the participants were migrant workers living alone, and were the sole breadwinners of their families. Five group experiential themes emerged relating to the experiences of the participants during the COVID-19 pandemic. Most participants reported significant psychosocial stress, but low suicide risk and robust coping mechanisms. They delayed seeking healthcare for non-COVID-19-related problems. Public hospitals were preferred over private hospitals due to cost constraints and prior experience of discrimination. Upward classism was observed as participants blamed the rich for the spread of COVID-19. Initial assumption that COVID-19 would only affect the rich was also reported. Free government testing and quarantine facilities assuaged their medico-psychosocial needs. Engaging in collective activities was a key stress mitigator. We highlight several important policy implications. Firstly, we underscore the importance of involving social workers to facilitate communication between healthcare providers and patients from vulnerable communities. This engagement can help minimize discrimination and promote equitable access to healthcare. Secondly, we emphasize the need for effective public health communication. Specifically, there is a need to address and alleviate concerns about the transmission of COVID-19 within hospital premises. Lastly, the research suggests that government initiatives aimed at fostering community participation should persist both during and after the pandemic.

Data are available from the Sri Jagannath Healthcare and Research Center Independent Ethics Committee (contact: The Secretary, Sri Jagannath Healthcare and Research Center - Independent Ethics Committee, Steel gate, Main road, near hero showroom, Dhanbad, Jharkhand - 828127 sweta.sjhrc@gmail.com) for researchers who meet the criteria for access to confidential data.

**Funding:** This study was commissioned by the Savitri Ghantasala Center for Health Equity, Samanjasa Foundation, Chennai, India. The funders had no role in study design, data collection and analysis, decision to publish, or preparation of the manuscript.

**Competing interests:** The authors have declared that no competing interests exist.

## Introduction

Homelessness remains a major issue in India [1]. According to the latest census, approximately 1.8 million people (0.15% of the total population) are homeless in India [2]. The key drivers of homelessness in India are migration from rural to urban areas, [3] disability, [4] unemployment, [5] and mental health problems [5]. Homelessness is a public health concern associated with numerous physical and mental health problems [4]. Persons experiencing homelessness (PEHs) have a higher risk of morbidity and mortality compared to the general population; they are also less likely to access healthcare [4].

The coronavirus disease (COVID-19) pandemic has resulted in 6,545,458 casualties as of 29 September, 2022 [6]. Governments worldwide imposed lockdowns to contain the virus, while racing to vaccinate their citizens.

Pandemics have a substantial impact on PEHs [7]. The lack of safe housing and the conditions in shelters place PEHs at an increased risk of contracting COVID-19, [7] and given the high prevalence of comorbid conditions in this population, they are more likely to develop severe disease [8]. After the outbreak of COVID-19 in India, the Government of India imposed a nationwide lockdown from 25 March, 2020 to 31 May, 2020, [9] and subsequently announced various pandemic relief measures for the poor [10]. However, despite the increased disease risk, most pandemic relief measures announced by the central and state governments of India do not make provisions for PEHs [11]. This may be due to the paucity of research focusing on the unique challenges faced by PEHs during the pandemic.

With news of overburdening of hospitals due to COVID-19 making daily headlines, fear regarding COVID-19 runs high. In this context, this study aims to qualitatively investigate the lived experiences of PEHs, and explore their challenges and coping strategies against COVID-19.

## Methods

Phenomenology is a philosophical approach to the study of experience, which seeks to understand and describe lived experiences [12]. Among the various approaches to qualitative research—such as phenomenology, grounded theory, discourse analysis, and narrative analysis—the phenomenological method was selected because we aimed to comprehend the lived experience of a particular phenomenon (enduring the COVID-19 pandemic and the resultant lockdown) by a specific group of people (PEHs). As understanding the lived experience involves an in-depth investigation into meaning-making, hermeneutic, existential, or interpretative phenomenology were considered appropriate. Among these approaches, interpretative phenomenological analysis (IPA) was adopted because this method analyzes the convergence and divergence of individual participants' experiences while simultaneously retaining their individuality.

IPA focusses on descriptions of a phenomenon as experienced by an individual, and their idiosyncratic interpretation of the experience, which is contemplatively interpreted by the researcher [12]. Videlicet, IPA is a double hermeneutic approach [12]. where the participant interprets the meaning of their experiences, and the researcher interprets these interpretations. A sample size of 3–12 is acceptable for IPA [13].

### Ethical considerations and trustworthiness

The study was conducted in accordance with the national guidelines and the Declaration of Helsinki. All procedures were approved by Sri Jagannath Healthcare and Research Center–Independent Ethics Committee, Dhanbad, India (No: SCR013/2020). This study ensures trustworthiness by employing several strategies. Credibility is upheld through prolonged

engagement with participants, while transferability is facilitated by providing context-rich descriptions. Dependability is maintained through comprehensive documentation of the research process, and confirmability is achieved via peer debriefing, enhancing the overall validity and rigor of the findings.

## Participants

The inclusion criteria were (1) age >18 years, (2) meeting the criteria of a "houseless person" as defined by the 2011 Census of India, [2] i.e., those "who do not live in buildings or census houses but live in the open on roadside, pavements, in hume pipes, under flyovers and stair-cases, or in the open in places of worship, mandaps, railway platforms, etc.", and (3) consenting to participate in the study and to publish their anonymized interview excerpts. The study was conducted with the help of the Savitri Ghantasala Center for Health Equity, Samanjasa Foundation, Chennai, India—a non-governmental organization (NGO) that engages with PEHs in Chennai and provides them various services including emergency interest-free loans, hot meals, medical checkups, and facilitating access to government schemes and facilities. They continued providing these services during the COVID-19-induced lockdown. Their case workers disseminated information about the study to PEHs who accessed their services. To build rapport with PEHs, a case worker arranged for the authors to meet the PEHs who had expressed interest in knowing more about the study at a location they are familiar with—the food stalls frequented by them—on two occasions. At these icebreaker sessions, the authors shared a meal with PEHs and engaged in casual conversation. Through purposive sampling, 10 PEHs were approached by the authors after the icebreaking sessions and provided detailed verbal and written information about the study; 9 of them (5 females and 4 males) gave written informed consent to participate in their respective mother tongues. Five participants gave consent to publish anonymized interview excerpts along with the translations; four participants gave consent to publish only translated excerpts. For consistency, only translated excerpts were used herein.

## Setting and data acquisition

To facilitate the flow of thoughts, individual face-to-face semi-structured interviews were conducted, between 14–25 September, 2020, at the same food stalls where the icebreaking sessions occurred. The interviews, which lasted 40–60 minutes, were audio recorded and concluded in one sitting. They were conducted in Telugu, Hindi, Tamil, Bengali, or Odia based on the participant's choice; between them, the authors are fluent in all these languages.

The participants were encouraged to expatiate their experiences. The key prompting questions were: (1) what was your opinion on COVID-19 when you first heard about it? (2) How did the pandemic affect you? (3) How have you been coping with the situation? And (4) have you needed a doctor since the pandemic began? The complete list of questions can be found in S1 File.

The participants were not offered any monetary compensation for participation in the study. However, at the conclusion of the interview, they were offered a gift card for a 1-month subscription to a popular over-the-top video service.

## Data analysis

The well-established process for IPA was followed for data analysis [12]. Briefly, the interviews were transcribed verbatim, and the transcripts were read while listening to the corresponding audio recording to get a general idea of the contents. The authors then contemplated on their biases and personal experiences relating to the study question and debriefed each other. Next,

the authors read the transcripts again and translated them into English. The authors subsequently read the translated versions and compared them with the originals to ensure that no change in meaning was introduced during translation. Next, all statements regarding each participant's experiences of or meaning attributed to the phenomenon of interest (experiencing the COVID-19 pandemic while being homeless) were coded into experiential statements. Related experiential statements were grouped together into subordinate themes, capturing the participants' remarks and the researcher's interpretation of the remarks. Subordinate themes were condensed into super-ordinate themes, capturing the essence of the participants' lived experiences. Finally, the super-ordinate themes from all transcripts were collectively abstracted by the authors into a set of group experiential themes and subthemes. Statistical analysis was conducted using Microsoft Excel 2016 (WA, USA).

## Results

All statements of each participant that underpinned the analysis are presented in the S2 File.

### Participant characteristics

The mean (±SD) age of the participants was 29.67±2.36 years. All the participants were born in other states and had migrated to Chennai for work. Their family stayed back in their native places, and they were living alone. All the participants were school dropouts and were the sole breadwinners of their families. They had their Ayushman Bharat (national free insurance scheme) card with them. The mean (±SD) daily wage in Indian Rupees was 317.59±167.06. Although some earned more than others in daily wages, they did not find work on all days of the year. They never tested positive for COVID-19. The participant characteristics are presented in Table 1.

### Exploration of the personal experiential themes

The following subsections present the personal experiential themes of each participant.

**PIN1.** PIN1 finds solace in the fact that her daughter is a good student, and that her sleeping in the street enables her to pay for her daughter's education. The COVID-19 pandemic

**Table 1. Participant demographic characteristics.**

| PIN[a] | Gender | Age (Years) | Number of Years of Formal Education | State of Origin | Occupation | Daily Wage in Rupees | Religion | Interview Language | Interviewed and Translated to English by |
|---|---|---|---|---|---|---|---|---|---|
| 1 | F | 32 | 4 | Andhra Pradesh | Construction laborer | 450 | Hindu | Telugu | Author 1 |
| 2 | F | 34 | 3 | Odisha | Housemaid | 83.33[b] | Hindu | Odia | Author 2 |
| 3 | M | 28 | 6 | Rajasthan | Watchman | 400[b] | Islam | Hindi | Author 2 |
| 4 | F | 26 | 4 | Tripura | Construction laborer | 100[b] | Islam | Bengali | Author 2 |
| 5 | M | 30 | 3 | Bihar | Construction laborer | 500 | Hindu | Hindi | Author 1 |
| 6 | M | 29 | 5 | Bihar | Construction laborer | 500 | Islam | Hindi | Author 1 |
| 7 | F | 30 | 7 | Kerala | Housemaid | 91.66[b] | Islam | Tamil | Author 1 |
| 8 | M | 27 | 7 | Odisha | Peon at an office | 333.33[b] | Hindu | Odia | Author 2 |
| 9 | F | 31 | 5 | West Bengal | Construction laborer | 400 | Islam | Bengali | Author 2 |

[a] Patient identification number;

[b] Paid monthly.

and the resultant lockdown enforced by the government rendered the laborers jobless and hence, she is worried about her daughter not being able to continue her education in the absence of funds. She anticipates self-guilt in case that happens. At the outset, she does not seem fearful of contracting the virus herself, as she dismisses it as if it were a foreign land phenomenon, but when it comes to her family, she exhibits palpable fear of transmission, particularly for the older folk. She, like many other participants, holds the rich accountable for bringing this crisis upon them. She seems to begrudge the fact that the rich could avail the best medical care while it remained out of bounds for people like her. Watching videos on YouTube and talking to her family over video calls help her cope with the ensuing distress. She mentioned her family multiple times during the interview indicating her profound attachment with her family, which probably keeps her going.

**PIN2.** PIN2's optimism of returning to normalcy hits a roadblock as she realizes that for housemaids, particularly those who are homeless, it is more difficult to get back to work even after the lockdown restrictions are lifted not only because their workplace is their employer's home, which augments the risk of transmission to the employer's family members, but also because PEHs are perceived to be more susceptible to contracting the disease. Being the sole breadwinner of the family, she fears the repercussions of contracting the disease on her family's future, so much so that she contemplates skipping meals as the place where she gets her food is crowded. She appears to care for her family deeply, while at the same time, she and her family take out the frustration of the pandemic on each other. This is probably their way of catharsis rather than animosity. One positive of this pandemic for her has been that now she regularly gets three meals a day, unlike earlier, thanks to an NGO. However, out of a deep-rooted sense of self-respect, she volunteers to wash dishes in return for the food.

**PIN3.** PIN3 comes across as a very responsible person who takes special care in following the COVID-19 safety guidelines so that he does not end up spreading the virus to others. He has sincerely followed various media outlets to obtain the necessary information about the disease, and exhibits good understanding of the different modes of transmission of the virus. He is a deeply caring individual who is more worried about the safety of others than of himself. He strives for human contact but he rigidly abides by the social distancing norms so that later he does not have to feel the burden of guilt if something bad were to happen to others due to him. He also wants to go home as he has not gone home for years, and his wife repeatedly urges him to. However, he shows restraint for the greater good, as going home involves the risk spreading the virus to the family members, and the risk of him losing his job. He is stressed that his wife does not see the bigger picture like he does. He enjoys having conversations with his friends and family via social media applications and hails technology for making it possible to communicate with them. He blames the rich for every COVID-19 related death—something that most participants have expressed in different ways.

**PIN4.** PIN4 like many other participants, lost her job during the pandemic, and her being homeless is an impediment in getting a job due to the stereotype of dirtiness associated with homelessness. Apparently, due to low self-esteem, she tolerates this kind of discrimination and abuse by the employer. She has a good awareness of COVID-19 safety protocols and follows them diligently. However, she has developed chest pain, for which she hesitates to get treated at public hospitals out of fear of transmission. She dismisses it as a minor risk to her health as compared to contracting COVID-19. Clearly, the way she takes precautions to avoid contracting the virus shows her desire to stay healthy. She says—"I am healthy and I want to stay that way". Nevertheless, due to lack of choices, she is unable to pursue treatment for her chest pain. She is reluctant to go to private hospitals due to her prior experience of being judged by the other patients at private hospitals. She seems to care for her self-esteem so much so that she is willing to wait out the pandemic for the treatment of her ailments instead of going to a private

hospital. Apart from loss of livelihood, the uncertainty about the future is a prominent cause of her anxiety.

**PIN5.** PIN5 too dismissed the possibility of contracting the virus in the initial phase of the pandemic. He has a sturdy consciousness of his own dignity and does not want to feed on freebies. This is visible from the fact that despite having a parental home, he chooses to sleep on the streets so that he can earn his food and live a dignified life. However, the pandemic has taken away the one thing that he prided himself upon, thereby rendering his life meaningless. The loss of livelihood has also led to the loss of the principles he stood by and lived for, as he is now compelled to take free food. He appears to be extremely dejected to see all the sacrifices he made for an honorable life going in vain. He thoroughly detests being dependent on others for food. He finds comfort in God/religion. Social media is the conduit for relief and coping. He also mentions experiencing discrimination at a private hospital—a common observation by some other participants.

**PIN6.** PIN6 is a construction worker who does not have a home but has two companions with whom he shares a corner on the street. He exhibits an emotional need for belongingness, as visible from the fact that he gravely misses eating and talking with his two companions who left for their native places after the lockdown was lifted. His inherent desire to belong and be an important part of a greater cause is also evident from the way he rejoices in participating in the engagement activities initiated by the Prime Minister of India. He goes through mixed emotions throughout the pandemic. On one hand, he is pessimistic about the whole situation, to the point that he has lost his faith in God. On the other hand, he feels optimistic about the future, hoping to become a Yoga teacher one day, thereby putting an end to his homelessness. He associates homelessness with humiliation, and having a home with dignity. Hence, he feels extremely grateful to the person with whose help he is able to learn yoga, through which he hopes to get rid of homelessness one day.

**PIN7.** PIN7 provides the unique perspective of a housemaid who experiences homelessness usually but has found shelter during the pandemic as her employer has let her move in to their house. She hopes that moving in would not only ensure income but also companionship. But to her dismay, she still feels lonely. This can probably be attributed to the socioeconomic class divide between her and her employer, and the resultant disconnect between them. The news of a fellow housemaid succumbing to the disease has aggravated her fears. She fears not only for her own safety but also for that of her family members who are financially dependent on her. Which is why she makes an extra effort in obeying COVID-19 safety guidelines like hygiene and social distancing. This is also one of the reasons why she hesitates to go to crowded public hospitals. She admits she was not a very devout person before the pandemic, but during the pandemic she became one.

**PIN8.** PIN8 comes across as a person with a positive outlook about things. Notwithstanding the discrimination he faces at his workplace on account of being homeless, and the consequent stereotypical association with dirtiness, he is grateful that he at least has a job during the pandemic and that he has been allowed to continue his job during the lockdown, unlike many who are not that fortunate. He sees the positive side of being homeless and attributes his immunity to diseases to his street-dwelling, which, according to him, makes his body "used to microbes". He is optimistic about the future and utilizes his free time to learn the English language, hoping to get a better job someday. Knowing about the free treatment and amenities provided at the government quarantine facilities gives him a passing sense of positivity about the much dreaded disease. However, he remains cautious of the disease and takes all precautions. His positive attitude is unequalled among all other participants. He finds it ironic that the rich people are practicing social distancing from the poor people like him, whereas, according to him, it was them who brought this disease to the country. He appears annoyed by the

unnecessary suffering inflicted on the poor because of the actions of the rich. Being aware of COVID-19 safety protocols, he prefers masturbation over risky sexual behavior as a coping strategy.

**PIN9.**   PIN9 is pragmatic enough to realize the threat of the virus and the uncertainty of the duration of the pandemic. Furthermore, based on the non-verbal clues expressed during the interview, it appears like she dislikes getting tested at Municipal Corporation health centers. Hence, she takes all precautions. She, like many other participants, has lost her livelihood due to the pandemic and the resultant lockdown. This has not only worsened her and her family's economic condition, but has also taken away her autonomy, as she highlights not being able to eat her choice of food, nor being able to complain about it, out of fear of being judged as ungrateful. In addition, her sense of dignity has been hurt; she says—"I am still surviving on doles. It is very frustrating". She, like most participants, holds the rich people responsible for the ongoing crisis. Substance usage is a stress mitigator for her. She admits she has been indulging in substance usage for a long time, and that it has increased during these stressful times. She comes across as one who resists change, as evident from the fact that she avoids trying a new substance, and that she initially feels reluctant to participate in the engagement activities initiated by the Prime Minister of India, which later, motivate her immensely and give her a sense of dignity and belongingness to a greater cause. Throughout her interview, she mentioned "family" repeatedly. Assuming that her family too might be struggling to get a source of income at her village, she sends them all the money she has. She chooses to remain homeless so that her family can get the comforts of a home, but under the current circumstances, she is worried that her family too may not be able to afford a home. Apparently, she cares for, and worries about, her family more than herself. A sense of hopelessness about the future could be observed from her choice of words and non-verbal cues.

## Exploration of the group experiential themes

Five group experiential themes emerged from data analysis (Table 2). The key takeaways along with representative excerpts that encapsulate each subtheme are presented herein.

**Attitude towards COVID-19.**   When India instituted a nationwide lockdown, most participants initially thought it would not affect them, and therefore, had nothing to fear.

> **PIN1:** ". . .somewhere in China if people are getting infected, why should we in India be so scared? The government is unnecessarily putting us all in a bad situation".

> **PIN5:** ". . .if you go to China, you will get it. Where do I have the money for that? I don't have money to go to my village itself. Rich people who travel here and there should be worried, not people like me".

A socioeconomic class divide was visible from the responses of the participants. The measures taken by the government to contain the pandemic were looked upon as unnecessary suffering inflicted on them because of the actions of the rich.

> *PIN8*: "Have you seen where I sleep? There is so much filth. Everyone says that you should be clean and keep the surrounding clean or you will get diseases. But in so much filth, I am very healthy. I have not missed even one day's work in so many years. My body can withstand anything. This corona is not going to harm me. Rich people who are worried about falling sick because of the rain should worry. They only travel to other countries and bring these diseases here and only they become sick because their bodies are not used to microbes. But government has imposed lockdown and we all have to suffer because of them".

**Table 2. Outline of group experiential themes and subthemes.**

| S. No | Group experiential themes | Subthemes |
|---|---|---|
| 1 | Attitude towards COVID-19 | Initial denial and lack of fear |
| | | Negative attitude towards the rich |
| | | Risk acceptance |
| | | Fear |
| 2 | Impact of COVID-19 on life in general | Isolation |
| | | Loss of livelihood |
| | | Difficulty in building trust with employers and co-workers |
| | | Positive experiences |
| 3 | Impact of COVID-19 on mental health | Anxiety |
| | | Loss of hope |
| | | Interpersonal stress |
| | | Sense of uncertainty |
| | | Anger against the rich |
| 4 | Coping strategies | Social media use |
| | | Participating in collective tasks |
| | | Religion |
| | | Sexual activity |
| | | Substance use |
| 5 | Barriers and facilitators to healthcare access | Reluctance to use public hospitals |
| | | Reluctance to use private hospitals |
| | | Free testing and quarantine facilities |

As more information about the disease started emerging, and the havoc it was wreaking all around them became apparent, the participants accepted their risk of contracting the infection and the consequences thereof. There was good COVID-19 safety awareness and compliance to guidelines among the participants.

**PIN4**: "I am healthy and I want to stay that way. I started washing my hands with soap after every three hours. I try not to meet anyone unless it is necessary".

**PIN7**: "I am constantly worried about getting infected. I don't want anyone to come near me. I have stopped taking my wages in cash. What if the virus is on the notes and I get infected because I used cash? I have started using BHIM (a government-run mobile payments service) to get my wages and to pay for anything. When my employer asked me to buy things for her house every day, I told her to get gloves for me. When I go to the shops, I don't go without the gloves and once I deliver the items, I throw away the gloves. How else can I be sure that I will not get Corona from the things I touch at the shops? If someone coughs or sneezes at the shops, I run away and go to another shop. I cannot afford to get infected".

**PIN9**: ". . .we have to accept the reality. Corona is not going away just because we wish it to go away. I have to wear a mask and if I get a fever I have to go to the (Municipal) Corporation health centre and get tested".

The fact that COVID-19 saturated the news cycle played a major role in the participants' acceptance of the disease and its risks.

**PIN3**: "The news, Facebook, everywhere you see, people are only talking about how many people are getting infected in the city. The threat of Corona has become very real".

With rising mortality due to COVID-19, the participants expressed fear of contracting the disease. The fear for their own safety seemed to stem from the fear of the economic consequences of their death for their families.

**PIN2**: "So many people are dying. There is so much fear in everyone. I am also scared. I am the only earning member of my family. If something happens to me, my family will have nobody to support them. The place I go to get my food is always crowded; I am so scared to get my food that I have even thought of skipping meals".

**PIN7**: "In the place where I work, I have to get the food items for the employers. Another housemaid I know got Corona by going to the market for food, and after getting Corona, she passed away. If the same happens to me, who will take care of my family? I try to keep my distance from others there, but I am scared for my safety. . ."

**Impact of COVID-19 on life in general.** Isolation occurred because of different reasons; however, it was a common theme among the participants. Being away from family seemed to be a major contributor.

**PIN3**: "I have not gone home for so many years, but I always thought I was only one train journey away. But they have stopped the trains now, and I am stuck here. I cannot meet anyone, nobody will come to meet me, and I have not spent time with another human being ever since this disease started spreading. . ."

**PIN6**: "There are two other people where I sleep, we usually eat together and talk to each other when we come back to our corner after the day's work. I thought they were my family away from my real family. But when the government lifted the lockdown, they have gone back to their families, and I am left with nobody".

**PIN7**: ". . .my employer asked me if I wanted to stay in her house and continue to work instead of going home and losing out on the money. I said I will stay. I thought I will have work, and have company to deal with the situation. But I felt lonely. They gave me food, shelter, and my wages. But they were not my friends or family. They can never understand what I am going through because they have never been in my situation".

For most participants, loss of livelihood was the biggest impact of the pandemic. The participants derived a sense of loss of autonomy from the loss of livelihood.

**PIN5**: "I ran away from home because I was a burden to my family. I came to the city and did some odd jobs. Whatever job I got, I did it wholeheartedly and ate whatever I could afford with my own money. I even started sending small sums of money to my family. I was fine with even sleeping on the streets, but I wanted to stand on my own feet. But ever since this Corona happened, I have not had any work. I am completely dependent on some kind hearted persons to provide me food. I hate my situation. I have to do what I ran away from home and slept on the streets to avoid doing; I am eating free food".

**PIN9**: "I have to eat whatever other people decide to give me. I don't like rice, but they (an NGO that distributes food) distribute only rice packets every day. If I say I don't want rice,

give me roti, I will be ungrateful. I miss making my own food, but I have no money. I sent whatever money I had to my family, thinking that they are in the village and it will be difficult for them to find work there, and the lockdown was only supposed to be a matter of two weeks. I thought I can survive on whatever the government gives for the two weeks and then earn again. But I was wrong. There has been no work for months now. I am still surviving on doles. It is very frustrating".

Once the lockdown restrictions were eased, the participants reported struggling to gain the trust of employers and co-workers. They tolerated abusive behavior due to low self-esteem.

**PIN8**: "I started going to work once the restrictions started easing. But nobody at the office comes near me. Nobody eats with me. Whenever I must touch anything, I must show the office staff that I am using sanitizer, only after that I am allowed to touch it. When I asked my supervisor why I must sanitize my hands so many times, but others peons don't have to, they said it was because I live on the street; it is not likely that I will maintain hygiene and social distancing on the street. I may be immune to Corona because of my street-dwelling, but others are not like me. It is a great thing that I got a job at all, and it is an even bigger thing that after Corona and the lockdown, I got to keep my job, so I do not complain much".

**PIN2**: "The restrictions are being lifted and I was hopeful of going back to work. But my employer told me not to come back. She is afraid I might give her family Corona. She does not want anyone from outside going to her house. Especially me, because she says I don't have a house so it will be impossible for me to take all safety measures. She told me she is afraid of letting me in the house".

**PIN4**: "There is not that much construction work going on now. But I thought I will do any work that I can find. But I am not able to find anything. I tried for a job at a small eatery, but the owner threw me out. He said I am shabby and if he hires someone who looks like he sleeps on the road, whatever customers he might get will also run away, because who knows what germs I will bring with me. It was very hurtful, but such is life, what can be done".

Most participants also reported some positives from the lockdown.

**PIN2**: "I have not gone a single day without food. Before Corona, sometimes I would get only one meal per day, but now, I get three meals daily from an NGO. But how long will I live on charity? I offered to wash their utensils, during lockdown, they refused. But recently they agreed. Now I can get meals and don't even feel like I am getting it for free".

They seemed to have invested in themselves during the lockdown, and were looking forward to positive changes in their lives once normalcy returned.

**PIN6**: "When I was a kid, my father taught me Yoga. I practice it every day to keep healthy. Especially since this lockdown started, I have not missed practice even a single day. I practice on the footpath where I sleep. One day a Policeman saw me practicing. He asked me if I wanted to become a Yoga teacher. I told him I would love to, but I don't have money, and this is the life I am destined for. The Policeman asked me if I have a smart phone. I told him yes. He took my phone and enrolled me in an online Yoga teacher training program. He paid for it. The program is in English, but the teacher repeats everything for me in Hindi. Every time he sees me, he asks me how I am doing in the program. I will be a certified Yoga

teacher soon, and I will be able to find a stable job and move out of the streets. I don't have to live this humiliating life forever, all thanks to that man's kindness which I can never repay in my life".

**PIN8**: "There was nothing else to do, so I started learning English on Youtube. When I go to get my food, I talk whatever I learned that day with the food distribution volunteers. They laugh at my mistakes, but they also correct me when I am wrong and help me practice small conversations. If I can talk English, I will be able to find a better job. I am looking forward to such a day".

**Impact of COVID-19 on mental health.** The participants reported feeling anxious about the spread of COVID-19 several times during their interview. The underlying cause of anxiousness varied—their own safety, safety of their loved ones, and fear of being an asymptomatic carrier.

**PIN1**: "My parents are old. I heard that old people cannot survive Corona. I am worried about them. Even after travel has been allowed, I decided not to go back home because I don't want to take the infection with me and give it to them. My father does not like sitting at home, I am so afraid that he will go out and get infected".

**PIN3**: "I am not afraid of Corona. I am strong, so I will not get affected by it. But the Prime Minister said on TV (television) that even if we don't get infected, we can give the infection to others. I am always conscious of how close I am to another person. I am worried if someone comes too close to me, what if I get Corona? I will not get affected, but that means I will not know I have it. Then if I touch someone else, and they get it and they die? I will never be able to forgive myself. Ever since this Corona started, I have not touched another human being. I crave for the human touch, it makes me very sad to not touch another person, but I am worried I might spread Corona to them. I don't want anyone to suffer because of me".

However, they demonstrated similar behavior as a result of the anxiousness—extreme social distancing, or even self-imposed isolation. The loss of livelihood and the resultant economic conditions seemed to have dampened the sense of hope of the participants.

**PIN1:** "I have no work. I sleep on the street so that I can send money to my daughter. She needs it to study. She is a very good student. I want to make her a doctor. But I have not sent any money home since the lockdown. Nothing is open, where will I find work? My daughter must pay tuition fees soon, or the tuition master (private tuition) will not let her attend classes. Corona does not even seem to end; I don't know when I can work and send money to my daughter again. If she fails because she was thrown out of tuition, I will never be able to forgive myself".

**PIN9**: "I don't have a roof over my head. I was sending all my money home so that they can have a better life, but now I have no money to send. If this continues, soon my family will also have to join me on the streets. I have no opportunities. I used to hope for a better tomorrow. Now I only pray that tomorrow is no worse than today".

They also expressed a sense of uncertainty about the future; it was a major stressor, which prevented them from feeling optimistic.

**PIN4**: "When will this Corona leave us? Will we ever get our life back? Only God knows. How can I make plans without knowing what tomorrow will look like? This uncertainty makes me very anxious".

Most participants reported significant interpersonal stress. The lockdown significantly impacted their mental health and tested the strength of their personal relationships.

**PIN2**: "My father is frustrated about being confined to the house, and whenever we talk, he takes it out on me. He is always picking fights with me over every small thing. I am also very much frustrated with the situation. But obviously I can't argue with my father, so I show my irritation to my mother".

**PIN3**: "My wife is constantly nagging me to come home. I still have a job, if I go home now, I will not have a job to come back to. My wife refuses to understand this, she just wants me to come home. I already miss my family; will I not go home if I could? And what about the risk of spreading the virus if I go home? I am tired of explaining all this. It is causing me a lot of mental strain".

Anger against the rich was expressed several times throughout the interviews. The participants held the rich responsible for the spread of COVID-19 in India.

**PIN1**: "The rich people knew they will bring Corona to India if they travel now, but still they travelled. Not just that, they lied to the government about their symptoms and spread the disease to innocent poor people who drive their cars, and work in their houses. Then they happily took the best medical care and left the poor people to fend for themselves".

**PIN3**: "I have seen the news how they (the rich people) took medicines to lower their fever just before coming out of the planes to escape the authorities. Some of them escaped from the quarantine facilities. What will happen if they spend 14 days in government quarantine? Will they die? They lied and spread the virus everywhere. These rich people are responsible for every single Corona death in this country".

**PIN8**: ". . .our country did not have this disease; it came from outside and the rich brought it here. They spread the disease and now they are acting as if we poor will give them the disease and want to be as far away from us as possible. If anything, we should be treating them like untouchables, not the other way around".

**PIN9**: "These (rich) people can't live without luxuries even if it means poor people will die because of them".

As the participants exhibited significant anxiety and depression symptoms, they were referred to an independent psychiatrist as mandated by the risk reduction policies of the Sri Jagannath Healthcare and Research Center–Independent Ethics Committee, Dhanbad, India. Upon evaluation and follow-up, the independent psychiatrist found that the participants did not meet the DSM-V diagnostic criteria for generalized anxiety disorder or major depression, and were determined to be at minimal risk for suicidality.

**Coping strategies.**   During the lockdown, most participants reported using social media to maintain contact with their families. It served as an effective conduit for social and emotional support.

**PIN1:** "I videocall my family over WhatsApp and talk to them for at least an hour every day. It is the bright spot in my day.

**PIN3:** "I work in the nights. In the morning I sleep. When I am awake, I message my friends and see their photos on Facebook. Sometimes I call them as well. My wife calls me on WhatsApp, and I see my parents and my son on video. I am happy to see them like this; they can be safe in the village and I also get to spend time with them. Technology has made it easier to tide over this crisis".

Most participants also reported participating in collective activities set by India's Prime Minister during his televised addresses to the nation to cope with the situation. Feeling like a part of something bigger than themselves helped the participants give meaning to their personal hardships and was an effective coping mechanism.

**PIN6:** "The Prime Minister used to come on TV and motivate us. I used to look forward to the next task he will give the country. When I clapped for the doctors, lit a candle, and did not venture out during the Janata curfew (the initial one-day lockdown in India on March 22, 2020), I felt collectively we will beat Corona. First time in many years, I felt like a part of this country, a part of a good cause. But now, the health minister comes on TV. He says something and goes. I do not understand much. Slowly that sense of belonging to a cause has vanished. I hope the Prime Minister comes back to motivate us again".

**PIN9:** "I was very reluctant to do all those things the Prime Minister asked the people to do on TV. But I started clapping for the doctors just like he said, because everyone around me was doing it and I did not want to be the odd one. But as I clapped, I became more enthusiastic. I felt like a part of something big and important. Then when he came on TV again, I looked forward to his task. My job loss and not knowing what will happen tomorrow seemed to be in the interest of this important cause of beating Corona. I felt like a soldier, fighting for myself, my loved ones, and countrymen. It has been a great motivator and source of strength".

The participants mentioned God several times during their interviews. Most participants turned to religion as a coping mechanism. Participants who were not devout became devout, and those who were already devout remained devout.

**PIN5:** "I have joined a prayer group on Facebook. Every evening, I light a lamp in front of God, and open Facebook. The group admins do live streaming of Bhajans (devotional songs). I forget all my troubles and sing along with them for one hour. It is the only time I feel that my life is not a waste".

**PIN7:** "I have only prayed for three years during my life, when I was living with my parents. I believe in God, but just did not see the point in praying. But ever since Corona started, I have been praying every day. I pray for the virus to be destroyed and our lives to go back to normal again. I saw a video where the Maulvi (religious scholar) said that if we all prayed together, the virus stands no chance. I thought why not give it a chance. Once I started praying, it gave me a lot of peace. I pray every day now".

Two participants reported losing faith.

PIN6: "I have been praying since I was 12 years old. I have not missed a single Friday prayer, but lately, I have been thinking, if God is inflicting so much suffering on innocent people, is he really fair? I don't feel like praying anymore".

Several participants reported indulging daily in masturbation, which was an effective stress mitigator.

> **PIN8**: "I don't think there is a better method to forget your troubles and sleep peacefully than daily sex. My wife is in the village, and we must maintain distance from strangers, so I masturbate every night, once it is dark enough and I am sure nobody can see me. It calms my nerves".

Some participants used substances as a way of dealing with their stressors. These participants reported using substances for a long time; however, they acknowledged that they used more frequently since the pandemic began. Substance use served to calm emotions.

> **PIN9**: "I have always used opium and tobacco. They take my mind away from my reality and keep my calm, even happy. But now I use them a lot. I have not tried anything new; I think it is better to stick to what your body is used to".

**Barriers and facilitators to healthcare access.** Most participants said they would not go to a public hospital even if they experience serious symptoms. The fear of contracting COVID-19 determined whether the participants sought healthcare.

> **PIN4**: "I can only afford public hospitals, but they are crowded. Everyone keeps saying that we must avoid crowds. Why go to the hospital and risk getting Corona. . .chest pain is more likely to be just gas. The odds that I am going to have a heart attack at my age are slim. But the odds that I will get Corona if I go to the hospital are high".

> **PIN7**: "If I go to a public hospital, I will have to be there the whole day, because the queue to see the doctors, and for tests, and everything else is so long. I don't have that kind of time; I have to work. Besides, the longer I stay outside, the higher my chances of getting infected with Corona; I don't want to take that chance".

The participants were reluctant to use private hospitals for their health needs due to cost constraints and the fear of being judged by the hospital staff and other patients. The notion that private hospitals were exploiting the COVID-19 pandemic for profit was expressed by many participants.

> **PIN1:** "Private hospitals are for the rich; people who live a hand-to-mouth existence like me cannot even think of them. They will gobble up the five lakhs insurance money in one day and ask us to get out the next day. It is coming in the news, didn't you see? They have increased the prices so much that even if you go to see them now with tooth pain, they will make you sell some body parts to foot the bill. They are making money on dead bodies".

> **PIN4**: "I don't like going to private hospitals. The hospital staff are okay, but the patients and their families look at people like me as if we are dirty. If I ever need to go to a hospital, I will wait till Corona is no longer a problem, and then go to a public hospital".

> **PIN5**: "I used the Ayushman Bharat scheme (India's health insurance scheme for the poor) at a famous private hospital. They treated my condition, but as long as I was there, they never looked at me like just another human being. I was always the guy who lived on the street. Nobody likes being looked at like that".

Most participants were appreciative of the free testing and quarantine facilities provided by the local government. Uptake of the free testing facilities was high among the participants due to the non-judgmental behavior of the staff.

**PIN2**: "I have been tested multiple times now. The Corporation (local government) officials took my phone number and regularly call me to find out if I am having any symptoms and if I need any help. Not once did I feel judged. Because I don't have to worry about paying for it, I go for testing whenever I get any symptoms".

The free food and beds at the quarantine facilities also appealed to the participants.

**PIN8**: "I know someone who also lives on the street who tested positive for Corona. The Corporation sent an ambulance to pick him up. He said he was taken care of very well; he was given three hot meals a day, and juice, tea, and coffee whenever he wanted. He had a clean bed, and he was treated for free. Ever since I came to know about the amenities at the quarantine facility, I have been thinking that getting infected with Corona may not be too bad; I can food and a decent bed to sleep on. But what if I get a severe attack, so I take all precautions and avoid getting infected".

## Discussion

This study explored the experience of the COVID-19 pandemic through the lens of PEHs in Chennai, India. The idiographic approach adopted by this study provides insights into the impact of the pandemic on the lives and mental health of a vulnerable section of society.

Previous studies have shown that pandemics trigger widespread mental health problems [14]. In this study too, all participants reported a myriad of psychological issues, which were largely attributable to the isolation, economic disadvantage, and loss of autonomy as a result of the national lockdown, rather than to COVID-19 itself. The associations between mental health and social isolation, [15] job loss, [16] and loss of autonomy [17] are well established. The interpersonal stress experienced by the participants may also be a result of the lockdown; however, we could not confirm the same, as exploring the interpersonal relationships of the participants prior to the pandemic was outside the scope of this study. This finding agreed with the findings of a previous study on college-going youth in India [18].

Denial and lack of fear of COVID-19 have been documented previously [19]; however, not at the initial stages of the pandemic, as is the case herein. The pandemic originated in Wuhan, China, and the initial cases in India were among those who had travelled abroad. The PEHs could not relate to the initial COVID-19 cases in India. Based on the demographics of the initial COVID-19 cases in India, the participants formed a stereotype of who will get affected by COVID-19, and opined that they were far removed from the stereotypical patient. Their initial denial and lack of fear were informed by this opinion. The rising number of cases among those who had never travelled abroad persuaded them to accept their risk. Acceptance gave rise to fear once persons they strongly identified with—co-workers and other homeless persons—succumbed to the disease. As a result of their being the sole breadwinners for their families, the participants were worried about the economic consequences for their families if they were to succumb to the virus, rather than their own safety.

The participants presented minimal suicide risk, suggesting that they had evolved robust coping mechanisms. Performing collective activities during the lockdown increased the sense

of belonging and the social engagement of the participants, which is known to be protective against depression and other mental illnesses [20].

The Government of India has used social and conventional media, celebrities, hoardings at prominent public places, etc. to spread awareness about COVID-19. It appears that the omni-channel communication efforts have been successful because the participants exhibited good awareness of COVID-19, and tried their best to comply with public health guidance; other studies have also found good levels of COVID-19 safety awareness and compliance with preventive measures in India, [21] thus supporting this notion. Similar to a previous study, [22] the pandemic exacerbated substance use in our participants. The government must accordingly create effective public health messages targeting drug use among PEHs and increase the capacity of its drug deaddiction program.

While most studies focusing on classism amid the COVID-19 crisis appertain to prejudice against the poor, [23, 24] our study found classism and resentment against the rich. The participants blamed the rich for the spread of COVID-19 in India. The frustration and hardships brought on by the pandemic and the resultant lockdown were attributed to the rich. Although another study reported on 'upward classism' in media portrayals of the rich in the context of COVID-19, [25] to the best of our knowledge, this is the first study to report 'upward classism' among members of the general public in this context.

Due to the high cost and prior experience of discrimination in private hospitals, the PEHs did not use private healthcare, and due to the fear of contracting COVID-19, they did not use public healthcare—this is consistent with previous studies conducted in other settings [26]. Instead, they preferred to wait out the outbreak and use public healthcare once the pandemic ends. A novel finding of our study is that the participants exhibited a positive attitude towards the government-backed quarantine facilities on account of them being free, as well as due to the provision of meals and beds along with medical attention. The quarantine facilities were perceived to address the medical and psychosocial needs of the PEHs. We previously reported on society's perception of profiteering by the healthcare workers during the pandemic [27]; the current study's results consistently establish this perception of private healthcare among the most vulnerable section of society.

The strength of our study lies in its thorough and comprehensive exploration of participants' lived experiences. Given the heterogeneity of our participants, our study offers a diversity of perspectives that enriches our analysis, providing depth and complexity to our understanding of the phenomenon under investigation. Our study holds significance in contributing to the broader understanding of the interplay among socioeconomic factors, public health, and the process of meaning-making.

A key limitation of our study is that the experiences of PEHs in other locations may differ from those in Chennai. Additionally, as Samanjasa Foundation ensured that PEHs who engaged with it accessed all government schemes available to them, we could not recruit PEHs without Ayushman Bharat cards—their lived experiences may differ from those with Ayushman Bharat cards due to a potential difference in their perceived access to healthcare. Thus, our findings cannot be generalized. Furthermore, active government and NGO involvement during the lockdown may have influenced our results. To comprehend the true aftermath, research when conditions return to pre-pandemic norms is essential. However, our study provides insights based on which future nomothetic studies to reveal the knowledge, attitudes, and practices of PEHs to protect themselves from COVID-19, and the difficulties faced by them amid this pandemic can be conducted. An idiographic study exploring the impact of the COVID-19 pandemic on the lives of the participants of this study post easing of restrictions and post vaccination rollout can enable a direct comparison of findings between the lockdown and post-lockdown periods and measure the impact of vaccination on this population.

## Policy implications

Although healthcare facilities are envisaged as safe spaces for all individuals to seek healthcare, in practice, bias against PEHs seems to be prevalent [28]; there is a need to appoint social workers to liaise between patients from vulnerable sections of society and healthcare providers at all hospitals, especially private ones. The social workers must act as effective advocates for patients, with the role possibly modelled after the Aboriginal Health Liaison Officers in Australian hospitals [29]. The protocols established by hospitals to prevent COVID-19 transmission in their premises and the importance of not delaying hospital visits for non-COVID-19-related problems must be publicized widely to allay fears and ensure that morbidity and mortality from treatable causes can be prevented [30].

The government must continue to organize collective activities to motivate the citizens and mitigate stress during the pandemic and must consider organizing such activities regularly beyond the pandemic as they seem to provide a sense of belonging to the PEHs—this may have a positive effect on the homelessness situation as a sense of belonging can facilitate reentry of PEHs into homes and society. Access to safe housing is an important determinant of health [4] In June 2015, the Government of India launched a scheme to provide housing for all Indian households by 2022 [31]. However, the scheme is unlikely to address the homelessness problem posed by migrant labor. In this study, the families of the participants lived in houses in their native villages while the participants had migrated to Chennai in search of work, and were experiencing homelessness. As migration is an important driver of homelessness in India, [3] any scheme addressing the homelessness problem must pay attention to housing for migrants. It is suggested that the mandate of the slum clearance boards (present mandate is to build and permanently transfer low-income housing to eligible beneficiaries) be expanded to operating low-income housing for migrant laborers on a subsidized rental basis. This may mitigate the homelessness problem driven by workforce migration and unaffordable rental costs. The number of homeless shelters must be increased, and for the duration of the COVID-19 pandemic, the government must strive to facilitate access to safe accommodation to all PEHs so that they can safeguard themselves against COVID-19.

## Conclusions

The present study has uncovered significant patterns within the lived experiences of PEHs in Chennai during the COVID-19 pandemic. These patterns, encompassing various thematic dimensions, highlight the multifaceted impact of the pandemic on this vulnerable population. The participants' attitudes towards COVID-19, initially characterized by denial and lack of fear, evolved into complex emotions including negative perceptions towards the rich, risk acceptance, and a palpable fear. The pandemic's broader implications on their lives were manifested through isolation, loss of livelihood, challenges in building trust within work environments, and unexpected positive experiences amid adversity. Our exploration of the pandemic's impact on mental health unveiled a range of responses, including heightened anxiety, loss of hope, interpersonal stress, uncertainty, and expressions of anger towards socio-economic disparities. Notably, the coping strategies employed by the participants spanned social media engagement, participation in collective activities, religious practices, and substance use. Furthermore, we illuminated the barriers and facilitators that shaped healthcare access in PEHs. While reluctance to use both public and private hospitals was observed, the provision of free testing and quarantine facilities emerged as significant facilitators in this context.

With the COVID-19 pandemic continuing to spread in India, and the state and central governments still not promulgating COVID-19-related policies directed at PEHs, they continue to suffer through the pandemic, without adequate access to services. Ensuring safe

accommodation can improve the physical and mental resilience of PEHs against COVID-19; we hope our findings trigger a policy dialogue to provide them directed COVID-19 relief.

## Supporting information

**S1 File. Prompting questions: List of key and ancillary prompting questions.**
(DOCX)

**S2 File. Appendix: Translations of all noteworthy utterances of each participant that underpinned the analysis.**
(DOCX)

## Acknowledgments

The authors would like to thank Gargi Kothari-Speakman of Savitri Ghantasala Center for Health Equity, Samanjasa Foundation, Chennai, India for facilitating rapport with the study participants.

## Author Contributions

**Conceptualization:** Laalithya Konduru.

**Data curation:** Laalithya Konduru, Nishant Das.

**Formal analysis:** Laalithya Konduru, Nishant Das.

**Investigation:** Laalithya Konduru, Nishant Das.

**Methodology:** Laalithya Konduru.

**Project administration:** Laalithya Konduru.

**Supervision:** Laalithya Konduru.

**Writing – original draft:** Laalithya Konduru, Nishant Das.

**Writing – review & editing:** Laalithya Konduru, Nishant Das.

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
