## [Editor Report · Decision Letter 0]

14 Nov 2022

PONE-D-22-30003Experiencing the COVID-19 Pandemic as a Homeless Person in Chennai, India: An Interpretative Phenomenological AnalysisPLOS ONE

Dear Dr. Konduru,

Thank you for submitting your manuscript to PLOS ONE. After careful consideration, we feel that it has merit but does not fully meet PLOS ONE’s publication criteria as it currently stands. Therefore, we invite you to submit a revised version of the manuscript that addresses the points raised during the review process.

ACADEMIC EDITOR: Be sure to:<ul> <li> 

Line 71 i.e. the operational definition of houselessness should be put up explicitly. The consenting procedures should be specified well ie verbal or written, and mode of language. Methods should have more clarity as to how PEH was approached ie through a civic or political body to gain their trust and confidence. Due to the rarity of these studies, it would be good if the consenting format may be a part of the submission (additional files). The ice-breaking sessions and conclusion of the interview i.e. the rights of these subjects in terms of time and sharing their experiences should be described as, were they offered free check-ups/food or monetary compensation for their narratives, should be described in better detail.

Line 603 all PEH in the study having the Ayushman Bharat card is a big limitation. Hence policy implications are limited as the benefit far outweighs the risk. The riskier are those who did not have the card.

Sociodemographic profile like education, family history, and reasons for living alone should be probed further. The impact of Covid 19 is looking better for this population due to heightened government activities like free food etc, so post covid restrictions pins would describe their situation better.

The age group is all in the sexually active period and given the conditions, it’s strange that they choose to stay alone. Chennai is the hub of AIDS & HIV, more details about their addictions and sexual habits would have added merit to the study.

A word on Covid 19 vaccination for this group would also be useful.

Study is good but the limitations are many and are to be discussed in more detail for benefit of readers. It qualifies to be a review article.

We look forward to receiving your revised manuscript.

Kind regards,

Sonali Kar, MD

Academic Editor

PLOS ONE

Journal Requirements:

"This study was commissioned by the Savitri Ghantasala Center for Health Equity, Samanjasa Foundation, Chennai, India and provided LK and NI with travel and food expenses and accommodation for the duration of the study. The sponsor played no role in the study design, data collection and analysis, decision to publish, or preparation of the manuscript."

---

## [Author Response · Author response to Decision Letter 0]

2 Dec 2022

We truly appreciate the constructive feedback and have made every endeavour to incorporate your feedback in our revised manuscript. We have provided point-by-point responses to your comments in the resubmission cover letter.

---

## [Editor Report · Decision Letter 1]

14 Dec 2022

PONE-D-22-30003R1Experiencing the COVID-19 Pandemic as a Homeless Person in Chennai, India: An Interpretative Phenomenological AnalysisPLOS ONE

Dear Dr. Konduru,

Thank you for submitting your manuscript to PLOS ONE. After careful consideration, we feel that it has merit but does not fully meet PLOS ONE’s publication criteria as it currently stands. Therefore, we invite you to submit a revised version of the manuscript that addresses the points raised during the review process.

ACADEMIC EDITOR:Thanks for revisions. Some minor revisions suggested. Kindly do needful

We look forward to receiving your revised manuscript.

Kind regards,

Sonali Kar, MD

Academic Editor

PLOS ONE

Journal Requirements:

Additional Editor Comments (if provided):

Have listed some more clarifications to be addressed

---

## [Author Response · Author response to Decision Letter 1]

19 Dec 2022

Thank you for your insightful comments. We have provided point-by-point responses to all your comments in the file named "Response to Reviewers".

---

## [Decision Letter · Decision Letter 2]

8 Aug 2023

PONE-D-22-30003R2Experiencing the COVID-19 Pandemic as a Homeless Person in Chennai, India: An Interpretative Phenomenological AnalysisPLOS ONE

Dear Dr. Konduru,

Thank you for submitting your manuscript to PLOS ONE. After careful consideration, we feel that it has merit but does not fully meet PLOS ONE’s publication criteria as it currently stands. Therefore, we invite you to submit a revised version of the manuscript that addresses the points raised during the review process.

We look forward to receiving your revised manuscript.

Kind regards,

Nabeel Al-Yateem, PhD

Academic Editor

PLOS ONE

Journal Requirements:

Reviewers' comments:

Reviewer's Responses to Questions

**Comments to the Author**

1. If the authors have adequately addressed your comments raised in a previous round of review and you feel that this manuscript is now acceptable for publication, you may indicate that here to bypass the “Comments to the Author” section, enter your conflict of interest statement in the “Confidential to Editor” section, and submit your "Accept" recommendation.

Reviewer #1: (No Response)

2. Is the manuscript technically sound, and do the data support the conclusions?

Reviewer #1: Partly

3. Has the statistical analysis been performed appropriately and rigorously? 

Reviewer #1: N/A

4. Have the authors made all data underlying the findings in their manuscript fully available?

Reviewer #1: No

5. Is the manuscript presented in an intelligible fashion and written in standard English?

Reviewer #1: Yes

6. Review Comments to the Author

Reviewer #1: Please note that I’m invited to review the second revision of the manuscript. I have not reviewed previous versions of this manuscript.

The authors have performed an important study about homelessness in India during the COVID-19 pandemic. There were nine participants which were carefully approached to participate in the study. The authors performed semi-structured interviews and analyzed data with an interpretative phenomenological analysis.

The authors should state that they performed a qualitative study (methods) and might describe some participant characteristics (results) in the Abstract. The results are well-presented, however, they should be shortened to meet the word limit. The implications should be written out rather than enumerated.

The introduction starts with the main topic ‘homelessness’. The authors succeed in briefly summarizing and clearly presenting the need for the study. The introduction section does not require revisions in my opinion.

The methods section should start with an introduction of ‘phenomenology’ (design) rather than ‘interpretative phenomenological analysis’ (data-analysis). Why did the authors chose to explore the phenomenon from an interpretative phenomenological perspective? The participants and setting sections are well-written. I do have two questions about the prompting. Is it possible to present all prompting questions in a box/table? And why did the authors not include questions about homelessness in the prompting? It now looks like that COVID-19 is the topic of the study instead of homelessness. I do miss a ‘trustworthiness’ paragraph summarizing the most important techniques to increase trustworthiness of results. It may be helpful to structure the Methods section according to the Standard for Reporting Qualitative Research reporting guideline (SRQR).

The results section needs no major changes in my opinion. The themes are illustrative and the stories are breath-taking. I would, however, like the authors to bring the terms such as 'individual themes' and 'master themes' in line with IPA according to Smith. Terms used by Smith et al are, amongst others, notes, emerging themes (may match individual theme), and patterns (may match master theme?). The case descriptions are very informative and suit a description of IPA-results.

The discussion section needs to be shortened. Please discuss 2, max. 3 topics instead of the current 6. Please provide ‘strenghts’ of the study before ‘limitations’. Maybe shorten the limitations to max. 8 lines. In the conclusion, more use can be made of the patterns found in the study data.

The manuscript is well formatted. English language use can be a bit more academic here and there. I also recommend formatting the tables in APA style. I recommend minor revisions.

Kind regards,

Dr. Niek Koenders

Senior researcher Radboudumc, the Netherlands

7. PLOS authors have the option to publish the peer review history of their article (what does this mean?). If published, this will include your full peer review and any attached files.

Reviewer #1: **Yes: **Dr. Niek Koenders

---

## [Author Response · Author response to Decision Letter 2]

12 Aug 2023

We have made every endeavor to incorporate the feedback provided by the reviewers. We hope the revised manuscript is now suitable for publication in your esteemed journal.

---

## [Decision Letter · Decision Letter 3]

20 Sep 2023

PONE-D-22-30003R3Experiencing the COVID-19 Pandemic as a Homeless Person in Chennai, India: An Interpretative Phenomenological AnalysisPLOS ONE

Dear Dr. Konduru,

Thank you for submitting your manuscript to PLOS ONE. After careful consideration, we feel that it has merit but does not fully meet PLOS ONE’s publication criteria as it currently stands. Therefore, we invite you to submit a revised version of the manuscript that addresses the points raised during the review process.

We look forward to receiving your revised manuscript.

Kind regards,

Nabeel Al-Yateem, PhD

Academic Editor

PLOS ONE

Journal Requirements:

Reviewers' comments:

Reviewer's Responses to Questions

**Comments to the Author**

1. If the authors have adequately addressed your comments raised in a previous round of review and you feel that this manuscript is now acceptable for publication, you may indicate that here to bypass the “Comments to the Author” section, enter your conflict of interest statement in the “Confidential to Editor” section, and submit your "Accept" recommendation.

Reviewer #1: All comments have been addressed

2. Is the manuscript technically sound, and do the data support the conclusions?

Reviewer #1: Yes

3. Has the statistical analysis been performed appropriately and rigorously? 

Reviewer #1: N/A

4. Have the authors made all data underlying the findings in their manuscript fully available?

Reviewer #1: No

5. Is the manuscript presented in an intelligible fashion and written in standard English?

Reviewer #1: Yes

6. Review Comments to the Author

Reviewer #1: First of all, I would like to thank the authors for the excellent response to reviewer(s). It was pleasant to read and finely structured. In addition, the points of interest were satisfactorily incorporated into the manuscript. Below are just a few minor comments worth processing.

Abstract: The abstract now contains all relevant information. Spelling and grammar needs a final check. E.g., the sentence "To explore the lived experiences of PEHs during the COVID-19 pandemic, a qualitative study-interpretative phenomenological analysis-involving semi-structured interviews of nine PEHs was conducted in Chennai, India, at food stalls frequented by them, between September 14-25, 2020." does not read pleasantly and seems grammatically incorrect. Perhaps the text could be modified to "We conducted a qualitative study to investigate the lived experiences of PEHs during the COVID-19 pandemic. Data were collected with semi-structured interviews in nine PEHs from Chennai, India, recruited at food stalls between Septembre 14-25, 2020. We analyzed the data with interpretive phenomenological analysis".

Introduction: No remarks.

Methods: The method section has improved tremendously. Certain choices, such as the choice of phenomenology, are now clearly justified. The credibility of the study is now easier to assess. Thanks for the modifications.

Results: The terminology (e.g., personal experiential themes) is now uniform and easier to understand.

Discussion: The arguments in lines 640-647 are interesting but speculative in nature. In my opinion, this text can be removed. The sentence "The pandemic exacerbated substance use, as reported by studies conducted globally." seems redundant and insufficiently substantiated. Please remove.

Conclusions: No remarks. The patterns are now well explained and summarized in a concise conclusion.

Language: More use could be made of 'we'-perspective instead of "the study" or some other third-person perspective. The researchers participate in the study as interviewers. Therefore, it seems appropriate to change, for example, "The study's exploration of the pandemic's impact" to "We explored the pandamic's impact." Please check the manuscript for use of 'we'-perspective where possible.

7. PLOS authors have the option to publish the peer review history of their article (what does this mean?). If published, this will include your full peer review and any attached files.

Reviewer #1: **Yes: **Niek Koenders

---

## [Author Response · Author response to Decision Letter 3]

4 Oct 2023

We have provided the responses to all specific reviewer and editor comments in the file titled "response to reviewers"

---

## [Decision Letter · Decision Letter 4]

17 Nov 2023

Experiencing the COVID-19 Pandemic as a Homeless Person in Chennai, India: An Interpretative Phenomenological Analysis

PONE-D-22-30003R4

Dear Dr. Konduru,

We’re pleased to inform you that your manuscript has been judged scientifically suitable for publication and will be formally accepted for publication once it meets all outstanding technical requirements.

Kind regards,

Nabeel Al-Yateem, PhD

Academic Editor

PLOS ONE

Additional Editor Comments (optional):

Reviewers' comments:

Reviewer's Responses to Questions

**Comments to the Author**

1. If the authors have adequately addressed your comments raised in a previous round of review and you feel that this manuscript is now acceptable for publication, you may indicate that here to bypass the “Comments to the Author” section, enter your conflict of interest statement in the “Confidential to Editor” section, and submit your "Accept" recommendation.

Reviewer #1: All comments have been addressed

2. Is the manuscript technically sound, and do the data support the conclusions?

Reviewer #1: Yes

3. Has the statistical analysis been performed appropriately and rigorously? 

Reviewer #1: N/A

4. Have the authors made all data underlying the findings in their manuscript fully available?

Reviewer #1: No

5. Is the manuscript presented in an intelligible fashion and written in standard English?

Reviewer #1: Yes

6. Review Comments to the Author

Reviewer #1: All comments have been processed properly. I thank the authors for all their efforts and the excellent manuscript.

7. PLOS authors have the option to publish the peer review history of their article (what does this mean?). If published, this will include your full peer review and any attached files.

Reviewer #1: **Yes: **dr. Niek Koenders

---

## [Editor Report · Acceptance letter]

22 Nov 2023

PONE-D-22-30003R4 

Experiencing the COVID-19 Pandemic as a Homeless Person in Chennai, India: An Interpretative Phenomenological Analysis 

Dear Dr. Konduru:

I'm pleased to inform you that your manuscript has been deemed suitable for publication in PLOS ONE. Congratulations! Your manuscript is now with our production department. 

Kind regards, 

on behalf of

Dr. Nabeel Al-Yateem 

Academic Editor

PLOS ONE